# Effectiveness of Resistance Training with the Use of a Suspension System in Patients after Myocardial Infarction

**DOI:** 10.3390/ijerph17155419

**Published:** 2020-07-28

**Authors:** Agata Nowak, Michał Morawiec, Tomasz Gabrys, Zbigniew Nowak, Urszula Szmatlan-Gabryś, Vaclav Salcman

**Affiliations:** 1Department of Physiotherapy, Jerzy Kukuczka’s Academy of Physical Education, 40-065 Katowice, Poland; zbinow@gmail.com; 2Uppersilesian Center of Medicine and Rehabilitation AMED, 40-514 Katowice, Poland; mojtrenerpersonalny@gmail.com; 3Department of Physical Education and Sport Science, Faculty of Pedagogy, University of West Bohemia, 301 00 Pilsen, Czech Republic; tomaszek1960@o2.pl (T.G.); salcman@ktv.zcu.cz (V.S.); 4Department Anathomy Faculty of Rehabilitation University of Physical Education, 31-571 Krakow, Poland; ulagabrys1957@tlen.pl

**Keywords:** comprehensive cardiac rehabilitation, percutaneous coronary angioplasty, ischaemic heart disease, myocardial infarction

## Abstract

The aim of the study was to assess the effects of resistance training with the use of a suspension system on exercise tolerance, evaluated through an exercise test, and the changes in selected echocardiographic parameters of patients after myocardial infarction. The study involved 44 males. The subjects were divided into two groups: Standard (20) and Suspension system (24). All the subjects had undergone an angioplasty with stent implantation. The standard and suspension system groups carried out a 24-day improvement program comprising 22 training units. Each session consisted of endurance, general stamina and resistance training. Instead of resistance training, the experimental group made multijoint exercises with a suspension system. Statistically significant changes in both groups were observed in the parameters of the echocardiographic exercise test, such as test duration (*p* = 0.000), distance covered (*p* = 0.000), MET (*p* = 0.000), VO2max (*p* = 0.000) and SBPrest (*p* = 0.013). Additionally, SBPmax in the suspension system group improved (*p* = 0.035). The echocardiographic test revealed significant improvement of Left Ventricular Ejection Fraction in both groups (SP group *p* = 0.001, standard group *p* = 0.005). The lipid profile test in the SP group revealed statistically significant improvement of TC (*p* = 0.003), HDL (*p* = 0.000) and LDL (*p* = 0.005). Training with the suspension system had a positive effect on the change of exercise tolerance level, left ventricular function and blood lipid profile.

## 1. Introduction

Both chronic and acute heart failure may appear as the first manifestation of coronary artery disease and occur with both impaired and preserved systolic function. Congenital heart defects (CHD) are therefore one of the main causes of chronic heart failure and are diagnosed in every second patient. One of the most common mechanisms leading to the onset or worsening of heart failure in patients with coronary artery disease is myocardial ischaemia [1,2]. It can occur in the course of acute coronary syndrome and lead to impairment of both systolic and diastolic left ventricular function. A consequence of myocardial infarction may be a scar with subsequent dyssynchrony and post-infarction aneurysm, which increases the risk of heart failure. The increased risk of heart failure in patients with CHD may also be associated with the coexistence of other cardiovascular diseases, i.e., hypertension, cardiomyopathy, valvular or congenital heart disease, pulmonary hypertension and other organ diseases, e.g., diabetes, renal failure gout hyperthyroidism or hypothyroidism [3].

The majority of the patients undergoing rehabilitation suffer from ischaemic heart disease after myocardial infarction treated with coronary angioplasty or aortic-coronary bypass surgery, as well chronic heart failure. Physical training starting after the hospitalisation stage [4] is the basic constituent of rehabilitation. This training consists of kinesiotherapeutic exercises, such as breathing, endurance training and resistance training [4,5].

The literature rarely touches upon research based on innovative methods of training in patients with cardiovascular diseases. Training with the use of a suspension system [6] is one of such methods. It involves exercises with two belts and application of the resistance of one’s own body weight. The suspension system consists of special Y-shaped straps made of durable polymer. Both arms are adjustable in length and ended with foot or hand grips. The handles are made in the shape of a roller and are covered with a rubber lining, ensuring a firm grip. In addition, there are canvas foot loops at both ends. The shoulder length is adjusted using metal zippers. The whole system is attached with a snap hook to the adapted construction. Below the hitch there is an adjustable strap, allowing the suspension length to be adjusted in relation to the attachment height. The person doing the exercise holds the handles in their hands while, on the other side, the straps are attached to the platform with a carabiner. This kind of training makes it possible to perform a large number of exercises that increase strength, flexibility, mobility and balance without the risk of injury [7]. The resistance in the exercises is adjustable by means of changing the distance between the exercising person and the device attachment point [5]. The ease of installation and transport makes the device much more attractive than traditional and much more expensive exercise equipment [6].

The concept of suspension exercises is based on three fundamental principles: Resistance relative to the vector of force, equilibrium and pendulum. The resistance to the vector of force is the ratio of the angle between the ground plane and the lever with the gravity force applied. The principle of equilibrium and pendulum is based on the starting position in relation to the point of attachment of the device [4].

The literature includes information about the effective influence of suspension systems on parameters such as strength and endurance [8,9,10]. The aim of such exercises is to apply progressive tension throughout the body and to adapt neuromuscular control [7]. This kind of exercise involves effort based on strength and endurance. Training with a suspension system is intrinsically safe, as the load is the patient’s own body weight. At the same time, this method is very effective [6,11,12]. The forced natural and ergonomic position during this training is its fundamental principle, and the benefit to patients at the same time [11]. Training with the use of a suspension system is aimed at eliminating an overload of the osteoarticular system despite the applied load. It counteracts the occurrence of negative movement patterns that cause pain, so-called postural re-education. Training in this form stimulates the stabilization of the spine (no translation of the vertebrae) and corrects the disturbed pelvic trajectory. These are the main goals of functional training in motor preparation. The foundation of this activity is maintaining the neutral position of the spine (lordosis and kyphosis and pelvic rotation) despite high shear forces. This is possible thanks to strong abdominal muscles and muscular balance. The system additionally enables joint movements in full range of their mobility with the use of stretching elements [11,13].

The scope of effort is focused on resistance training and, unlike traditional forms of training, makes it possible to perform complex exercises in many planes. Parameters such as elasticity, equilibrium, balance and, above all, muscle strength, are developed simultaneously. The system enables seamless transition from endurance-based exercises with the use of straps to dynamic elements of the training, thus reducing the risk of injury to a minimum. Parameters such as trunk stability and proprioception are significantly improved by stimulating the neuromuscular system, which, in turn, is extremely important particularly in case of the elderly [6,14,15]. Training with the suspension system as a functional and stabilization training uses a different biomechanical mechanism based on the improvement of central stabilization based on muscle timing than traditional resistance training. It corrects the position of the pelvis and spine despite the fact that the movement takes place around the perimeter. Working in a closed kinematic chain makes it possible to limit and counteract the shear force acting on the joints. In contrast, traditional strength training is based on exercises in an open kinematic chain. Training using the suspension system as a stabilization and functional training improves such parameters as flexibility, stability, balance and the main parameter of muscle strength in a coherent and simultaneous manner. In addition, the choice of load is adjusted using the starting position, allowing safe adjustment of the intensity of the exercise.

To date, there have been no scientific reports on the possibility of using suspension training as an alternative to traditional strength and endurance (resistance) training in cardiac rehabilitation. The study made it possible to assess the value of this training as a modern therapeutic method applicable to patients after myocardial infarction. The aim of the study was to determine the usefulness of training with the use of a suspension system as an alternative to traditional strength and endurance training in cardiac rehabilitation. The research questions were as follows:Does training with the use of a suspension system change exercise tolerance assessed through an exercise test in patients after myocardial infarction compared to the control group?Does training with a suspension system compared to the control group cause changes in selected left ventricular echocardiographic indicators in patients after myocardial infarction?Does modified resistance training with the use of a suspension system change, similar to standard training, the blood lipid profile of patients after myocardial infarction?

## 2. Materials and Methods

### 2.1. Participants

The study involved 47 people: 44 men and 3 women. Due to the low number of women, only the results of a group of men were analysed. The patients were randomly assigned (random selection) based on two therapeutic strategies: 

**Standard group**: Patients subjected to improvement based on the recommendations of European Society of Cardiology (ESC, Table 1).

Suspension system group (SP group): Patients also trained in accordance with ESC recommendations. Instead of traditional resistance training (rowing machines, elliptical trainers, steppers), patients in the SP group carried out training with the use of a suspension system (Figure 1).

In both cases (standard and SP group), trainings were conducted in groups not individual.

Descriptive characteristics of the subjects (Table 2, Table 3, Table 4, Table 5, Table 6 and Table 7).

There was no significant difference in the age of the tested patients.

There was no significant difference in the body mass and body mass index (BMI).

Ischemic disease and myocardial infarction as well as hypertension were dominant in both studied groups.

All subjects had undergone an angioplasty with stent implantation.

The highest percentage of patients in both analysed groups had an angioplasty and one implanted stent.

Due to the risks associated with the introduction of the new training method and intense loads applied, only the patients who achieved the best result (≥7 MET or 100 W) at or above the target heart rate without any cardiovascular problems during the initial exercise test qualified for the study.

Study inclusion criteria:Uncomplicated myocardial infarction,Duration since last cardiovascular incident not lower than two months and not greater than six months,Complete revascularisation,Exercise test result ≥7 MET/100 W,Left ventricular ejection fraction EF ≥ 50%,Similar body weight in the 60–80 kg range (narrow range, homogeneous group, similar load),Height in the 170–180 cm range,Consent to participate in the study.

Main study exclusion criteria:Recent myocardial infarction–less than two months from the occurrence,Duration from myocardial infarction greater than six months,Incomplete revascularisation,Coronary artery bypass grafting,Not regulated hypertension,Unstable ischaemic heart disease,Arrhythmia and conduction disorders,Incomplete medical records,Left ventricular ejection fraction EF < 50%.

Additional criteria for the exclusion from the study included conditions, which, in the opinion of the qualifying physician, prevented the participation in the study:

Thromboembolic disease, metabolic disease, diagnosed cancer, diseases of the central or peripheral nervous system, varicose veins, degenerative disease of the peripheral joints and spine, advanced peripheral vascular atherosclerosis, age ≤ 75.

### 2.2. Experimental Procedure

Both the standard group and the SP group were subjected to a 24-day improvement programme, which included 22 training units (2 days for initial and final tests) performed 5 times a week following ESC Standards (a detailed training program is presented in the Table 8, Table 9 and Table 10). Throughout the entire research procedure, the patients were supervised by medical personnel consisting of a physiotherapist and a cardiologist.

Arterial pressure and heart rate were measured before, during and after each training. The level of perceived exertion was monitored by means of the 20-degree Borg scale. Training intensity was adjusted on the basis of training heart rate. It initially amounted to 60% of heart rate reserve and was increased by 10% for every five units to finally reach 80% of heart rate reserve, which represented, at maximum, the 14th degree of perceived exertion as assessed on the basis of the Borg scale.

In the SP group, the training was carried out following a plan prepared by the same trainer and at the same time (Figure 2). Each exercising person had his heart rate measured continuously by means of a heart rate monitor.

The training unit with the use of a suspension system included four multijoint exercises:

1. knee bend, 2. lunges, 3. pulling handles to the chest, 4. shoulder lift.

Each exercise was carried out in 3 series of 12 repetitions (3 × 12). The series were separated by 90-s intermissions. The exercises were carried out at a rate of 5 s per repetition, with a 3-s eccentric phase a 2-s concentric phase. The duration of a series of each exercise was 60 s (12 × 5 s). The entire duration was 30 min. To maintain constant load, the exercising persons stood at designated exercise positions at a respective distance from the attachment point of the suspension system. The length of strap arms was adjusted (exercises 1 and 2: 190 cm, exercises 3 and 4: 170 cm). The devices were hooked at a height of 2 m.

The tests were made with the use of professional suspension systems by TRX.

Starting position and movement:Knee bend. Starting position: Standing upright facing the straps, feet spread at the width of shoulders, handles held in hands, elbow joints bent at right angles. Movement: Bending lower limbs in knee and hip joints to a right angle.Lunges. Starting position: Standing upright with the back facing the straps, handles held in hands, elbow joints bent at right angles, feet spread at the width of shoulders. The movement consists of bending the lunged lower limb to a right angle in the knee joint and alternating the lunged lower limb.Pulling handles to the chest. Starting position: Standing upright facing the straps, handles held in hands, feet close to each other, shoulders at right angles relative to the torso, elbow joints upright. The movement consists of pulling the shoulder blades to each other, bending the elbow joints and pulling the handles to the chest.Shoulder lift. Starting position: Standing upright facing the straps, handles held in hands, feet close to each other, shoulders at right angles relative to the torso, elbow joints upright. The movement consists of lifting the arms straight in the elbow joints to an angle of 180 degrees.

Internal forces in the general centre of gravity of the human body were calculated for the four types of exercises. The figures below illustrate the exercises. The figures include diagrams of internal forces, i.e., bending moments and axial forces, as determined by the method of displacements (Figure 3, Figure 4, Figure 5, Figure 6 and Figure 7).

The table below (Table 11) shows muscle and spine loads during exercises, expressed in kilograms. The analysis used the 100% LPM test. It is a very simple and often used in sport and recreation. The optimal considered utilization is linked to the purpose of the training and the method to be used. Depending on the goal set, follow the equivalent of repeating the results of the exercises. The test consists in choosing the maximum direct selection to allow the preferred number of repetitions (Table 12). There was no need to convert the values available to obtain power compliance.

The following was carried out before commencing the training program and immediately after its completion: 

Electrocardiographic exercise test on a treadmill (Bruce protocol). The following was measured:

Test duration (min), distance covered (m), energy cost (MET), heart rate at rest and maximum (BPM), systemic blood pressure at rest and maximum (mmHg), criteria for ending the test (physiological: Submaximum heart rate, i.e., 85% of HRmax determined on the basis of the following formula 208−0.7 × age or fatigue; pathological: Stenocardial pain, ST segment, and T-wave changes, rhythm and/or conduction disorders, blood pressure increase above 250/120 mmHg), maximum oxygen uptake VO2max.

Two-dimensional ultrasound heart test, measured parameters:-LVEDD–Left ventricular end-diastolic dimension (norm: 35–57 mm)-LVESD–Left ventricular end-systolic dimension (norm: 22–40 mm) (Nowak 2006)-LVEF%–Left ventricular ejection fraction (norm: >50%)-LVM–Left ventricular mass -LVESV–End-systolic volume (ml) (norm: women 19–43 ml; men 22–58 ml) as per the following formula: LVESV= 7/(2.4 + LVESD) × (LVESD)-LVEDV–End-diastolic volume (ml) (norm: women 56–104 ml; men 67–155 ml) as per the following formula: LVEDV = 7/(2.4 + LVEDD) × (LVEDD)-LVSV–Ejection volume (ml) (norm: 75–100 ml) as per the following formula: LVSV = LVEDD − LVESD -LVM–Left ventricular mass (g) (norm: women 66–150 g; men 96–200 g) as per the following formula: LVM = 0.8 (1.04 × [(LVEDD + IVS + LVPW)^3^ − LVEDD^3^]) + 0.6

where

-IVS–intraventricular septum dimension-LVEDD–left ventricular end-diastolic dimension-LVPW–left ventricular posterior wall dimension -LVMI–left ventricular mass index (g/m^2^) (norm: women 44–88 g/m^2^; men 50–102 g/m^2^) calculated based on the Devereux formula [12]: LVMI = LVM/BSA

whereLVM–Left ventricular mass BSA–Body surface area as per the formula by Dubois & Dubois (Dubois & Dubois 1916).


Blood lipid profile test. Measured parameters:-Total cholesterol (TC) (mg/dL) (norm: < 190 mg/dL),-High-density lipoproteins (mg/dL) (norm: ≥40 mg/dL),-Low-density lipoproteins (mg/dL) (norm: ≥115 mg/dL),-Triglycerides (TG) (mg/dL) (norm: ≤ 150 mg/dL)

The study was carried out in an analytical laboratory by a qualified person (5 ml blood was taken for analysis).

### 2.3. Data Analysis

The Shapiro–Wilk normality test and the Brown–Forsythe variance homogeneity test were used to verify the assumptions of parametric tests. A parametric Student’s t-test was also performed for dependent variables whose distribution is in line with normal distribution, and a nonparametric Wilcoxon paired order test was performed for dependent variables whose distribution was not aligned with normal distribution. A Student’s t-test was also performed for independent variables whose distribution was aligned with normal distribution, and its nonparametric equivalent. Mann–Whitney’s U-test was performed for independent variables whose distribution was not aligned with normal distribution. Statistica 12 (StatSoft, Kraków, Poland) software was used in the study. The assumed level of significance was *p* ≤ 0.05.

## 3. Results 

### 3.1. Electrocardiographic Exercise Test

Table 11 shows the results for the two test groups of the treadmill exercise test as per the classical Bruce protocol.

In both rehabilitated groups (SP and Standard), significant improvements in exercise tolerance in terms of test duration, metabolic equivalent (MET) and VO2max were observed. A cross-group analysis of changes (delta) revealed significant differences in the metabolic equivalent (MET) and blood pressure at rest (Table 13).

### 3.2. Echocardiographic Test

Compared to the initial test, both rehabilitated groups showed a significant increase in the left ventricular ejection fraction. Whereas positive changes were also observed for the other assessed echocardiographic indicators, they were not statistically significant (Table 14).

### 3.3. Blood Lipid Profile

An improvement in lipid profile was observed in both rehabilitated groups.

The greatest and significant changes were found in the SP group. Whereas the changes of other indicators were positive, they had no statistical significance (Table 15).

## 4. Discussion

The results of the study point to positive changes in terms of the improvement of patients’ after myocardial infarction due to the application of a nonstandard method of resistance training, i.e., training with a suspension system. It is likely that such studies have never been carried out on cardiac patients, as trainings with the use of such a system are primarily carried out in fitness clubs and involve healthy individuals.

The analysis of the authors’ own research revealed that training with the use of a suspension system is not only a well-tolerated and effective method of resistance training, but it is a primarily safe and appealing proposition, which can be included into the existing methods of the second stage of complex cardiological rehabilitation. The availability and low costs of the training device in comparison with devices used in standard resistance training–elliptical trainers, rowing machines and steppers–is also an important benefit of this method. Moreover, a suspension system used for training is also an easily accessible tool (it can be installed at home, in the garden or in a park) for the continuation of safe activity in the third stage of cardiac rehabilitation.

### 4.1. Electrocardiographic Exercise Test

The obtained results revealed a significant improvement in exercise tolerance, as evaluated on the basis of an electrocardiographic exercise test results relative to the results obtained before the improvement program. After 24 days of rehabilitation, a significant increase in the test duration and distance covered, as well as MET and VO2max, were observed in both the SP and Standard group.

Maximum oxygen uptake (VO2max) is the basic indicator of endurance, particularly in case of prolonged effort and, at the same time, an indicator of cardiovascular performance. A maximum oxygen intake of 10 mL/kg/min represents severe heart failure. The minimum level of physical activity assessed via VO_2_max amounts to 40 mL/kg/min (11 MET). For a person with a sedentary lifestyle, VO_2_max amounts to approximately 30 ml/kg/min, which corresponds to MET of 8.5. The results of the study revealed a significant increase in VO_2_max for both trained groups, SP and standard (34.34 vs. 39.43 ml; *p* = 0.000 and 34.21 vs. 44.52 mL; *p* < 0.001, respectively), which demonstrates that a properly planned and implemented rehabilitation program carried out on a continuous and systematic basis leads to a significant improvement in physical performance of patients. Similar conclusions were also reached by Yang et al. [16], Guazzi et al. [17], Adams et al. [18] and Balady et al. [19], thus proving the positive effects of cardiac rehabilitation on the improvement of spiroergometric indicators of physical performance in patients with heart failure and after acute coronary incidents. A study by Jelinek et al. [20] also confirmed that a six-week cardiovascular rehabilitation program in patients after PCI is beneficial for physical performance, cardiopulmonary function and the autonomic nervous system, which modulates heart rate. MET (metabolic equivalent of a task) is another indicator measured during an electrocardiographic exercise test. According to Myers et al. [21], peak exercise capacity measured in METs is a very good predictor of the risk of death in both cardiovascular patients and healthy individuals. The authors’ own research demonstrated that both training with the use of a suspension system and standard training significantly improved MET (8.93 ± 1.22 vs. 9.96 ± 0.96; *p* = 0.000 and 9.99 ± 1.08 vs. 11.91 ± 1.86; *p* = 0.000, respectively).

A favourable increase in MET due to a cardiovascular rehabilitation program in stage II was also demonstrated in a retrospective 10-year analysis of 20,671 patients, regardless of the initial level of exercise tolerance [22]. A similar effect was observed in the assessment of the effectiveness of hybrid rehabilitation of 125 patients with coronary artery disease treated with the use of the interventional method. After the completion of a five-week improvement programme, a significant increase in exercise tolerance, and thus, in MET (7.86 ± 2.59 vs. 8.88 ± 2.67, *p* = 0.000), was observed [23].

Distance covered on a treadmill was another indicator of performance measured during the exercise test. Patients who trained with a suspension system, as well as those following a standard program, achieved a significant improvement, i.e., the duration of their final test was longer by about 2 min (*p* < 0.000) and the distance covered was greater by more than 100 m (*p* < 0.000). The obtained results are in line with the results of other authors [23,24].

Proper adaptation to physical effort and, at the same time, increased exercise performance, is manifested by slower heart rate at peak and at rest. This is affected by reduced activity of the autonomic nervous system. This mechanism, as well as the excitation of the vagus nerve, also lead to a decrease in both systolic and diastolic pressure [24].

In the authors’ own study, a significant decrease in resting heart rate was obtained (70.1 vs. 66.25 bpm; *p* < 0.013) in a group undergoing training according to ESC standards. A decrease in systolic pressure at rest was significant in both groups—Standard (129.75 vs. 122.5 mmHg; *p* < 0.016) and suspension system (122.92 vs. 114.17 mmHg; *p* < 0.013). The differences are probably attributable to the greater intensity of resistance training, which included exercises on elliptical trainers, rowing machines and steppers, compared to the SP group, where the training was less intense, and the load was applied by means of the patients’ own body weight. As far as blood pressure peaks are concerned, a significant difference was observed in the SP group in case of maximum systolic pressure (*p* = 0.035). Even though significant differences were observed, it appears that this result can be attributed to a random occurrence considering that pressure values were very similar in the SP and Standard group. The exact mechanism of lowering blood pressure as a result of regular physical activity is not fully understood. This is certainly due to the beneficial multidirectional impact on the physiological mechanisms of the circulatory system. There is probably a decrease in peripheral resistance and cardiac output, as well as a decrease in sympathetic tone.

Similar results, i.e., a significant decrease in heart rate at rest and an increase in maximum heart rate mainly related to the extension of the test duration, were also obtained by other authors [23,24,25].

### 4.2. Echocardiographic Test

Left ventricular indicators relevant to the effectiveness of the improvement program were evaluated. Even though both rehabilitation groups (SP and Standard) revealed positive changes, the only significant change involved left ventricular ejection fraction (LVEF). The changes were observed in the following: LVEDD, LVESD, LVESV, LVEDV, LVSV, LVM, LVMI. In the case of LVEF, however, both the SP and standard group revealed significant differences that indicated the improvement of left ventricular function (52.29 vs. 53.33%, *p* = 0.001 and 53.30 vs. 55.30% *p* = 0.005 respectively). It can only be presumed that in subsequent studies after the lapse of another three or six months, differences would be revealed to be significant in relation to the remaining parameters.

The 22-day training period was quite short to expect significant changes in the left ventricular hemodynamics, which was confirmed by the results obtained for both groups. Nevertheless, the substantial increase in LVEF shows an improvement in left ventricular contractility as a result of the training program. As this improvement was observed in both groups, it can be presumed that both the suspension system training and the standard program proved to be equally effective. On the other hand, it needs to be emphasised that the effects of physical activity on the structure and functions of the left ventricle in patients after myocardial infarction have yet to be clearly determined [26]. The reason may be attributable to, among others, the differences in the methodology of the research conducted by different authors, including factors such as the selection of patients, type and extent of myocardial infarction, age of patients, observation period and measuring methods. Although the findings of increased ejection fraction in our study do not correlate with LVESV and LVEDV changes, the sample size was relatively small. Also, the EF has limited value as a tool to accurately measure left ventricular function that potentially can cause bias.

Theoretically, multiple parameters could influence changes in LVEF, including:The use of beta-blockers that decrease wall stress and reduce heart rate,The use of ACE inhibitors that decrease afterload,Physical training that combines multiple parameters, including lower heart rate at rest, modulation in adrenergic activity, lower left ventricle wall stress, increased venous return and modified tension of arterial wall that leads to reduced afterload.

Left ventricular ejection fraction is an indicator of global myocardial contractility and is a key parameter determining the condition of patients after myocardial infarction [27]. It also reflects the effectiveness of comprehensive cardiovascular rehabilitation, as evidenced by the study by Doimo et al. [28].

A six-month observation by Belardinelli et al. [25] revealed that LVEF was a significantly differentiating indicator for patients. The authors demonstrated that for active individuals, there was a significant increase (52.3 vs. 57.3%, *p* < 0.000), which is consistent with the changes observed in their own analysis. Fahreen et al. [29] observed that at the end of a six-week improvement program of rehabilitating patients after myocardial infarction, in the group of combined resistance and aerobic training there was a significant improvement in the Left Ventricular Ejection Fraction (45 vs. 55%; *p* = 0.029 and 45 vs. 50%; ns respectively).

They attributed the increase in the left ventricular ejection to the improvement in endothelium and autonomic function due to resistance effort in combination with strength and endurance training. Chrysohoou et al. [30] reached similar conclusions in the case of subjects with chronic heart failure, however.

Haddadzadeh et al. [27] examined 42 patients with coronary artery disease and assessed the effects of a 12-week rehabilitation program. LVEF significantly improved (46.9 vs. 61.5%; *p* = 0.001) in relation to control subjects with no exercise (47.9 vs. 47.6%; ns). Sadeghi et al. [26] observed that after eight weeks of cardiac training in patients with left ventricular dysfunction, there were no significant changes in LVESD (38.91 vs. 38.09 mm; *ns*) and LVEDD (54.63 vs. 53.86 mm; *ns*). According to Nowak et al. [24], left ventricular mass (LVM) in patients over 40 years of age is an independent prognostic marker of coronary mortality. Left ventricular mass index (LVMI) is another parameter used to assess the extent of left ventricular hypertrophy. The results concurrent with the authors’ own research were provided by Nowak et al [24], who noted an insignificant decrease in weight and left ventricular index (210.64 vs. 207.86 g/m^2^; *ns* and 110.82 vs. 109.86 g/m^2^; *ns*, respectively). As the author himself pointed out, the lack of significant changes can be attributed to the short duration of observation. Pitsavos et al [31] analysed the effect of four months of physical training on the cardiovascular system in middle-aged men with normal or medium elevated blood pressure without CHNS symptoms, and also observed a significant decrease in LVM and LVMI in the training group (225.10 vs. 181.87 g and 118.80 vs. 96.10 g/m^2^, respectively; *p* < 0.05), while the increase in the group without physical training amounted to LVM: 227.73 vs. 231.13 g; *p* < 0.05, LVMI: 115.94 vs. 117.52 g/m^2^; *ns*.

The effects of physical training applied in the second stage of rehabilitation on myocardial function have not been clearly explained. Most studies, similar to the authors’ own studies, were unable to demonstrate a significant effect of training on the morphological and functional parameters of the left ventricle naturally, except for LVEF. It needs to be pointed out that both methods of training have impacted the increase of LVEF, which is considered by many researchers as one of the key prognostic parameters in patients after myocardial infarction [27].

### 4.3. Blood Lipid Profile

Even though lipid levels are determined on a hereditary basis, factors related to lifestyle (environmental factors), such as a proper diet and regular physical activity, play a significant role in keeping them low. Regular physical activity is one of the elements that contribute to the decrease of the LDL fraction, increase of the HDL fraction and reduction of triglyceride concentration. On the basis of the results obtained in the study, it was found that the applied improvement programme with the use of training with a suspension system had a positive effect on changes in lipid profile. Moreover, in case of the SP group, the changes were significant (total cholesterol *p* < 0.003, LDL *p* < 0.001, HDL *p* < 0.005, TG–positive change but ns), whereas, in the Standard group, the direction of the changes was also positive, but no significant improvement was observed. Belardinelli et al. [25] assessed the effects of physical training on changes in lipid profile and showed significant changes in total cholesterol, LDL fraction (235 vs. 212 mg/dL and 148 vs. 131 mg/dL, respectively; *p* < 0.001) HDL fraction (34 vs. 39.2 mg/dL; ns,) and triglycerides (178 vs. 155 mg/dL, *p* = 0.02). For the group of patients with no physical activity, after half a year, the researchers noted an increase in total cholesterol (225 vs. 255 mg/dL, *p* < 0.001), triglycerides (181 vs. 189 mg/dL; *p* < 0.001) and LDL fraction (138 vs. 148 mg/dL, *p* < 0.001). Other studies that analysed patients for a duration of six months after angioplasty with the first and second stage of rehabilitation demonstrated a substantial decrease in total cholesterol, LDL and triglycerides [24].

### 4.4. Limitations of the Study

Apart from the fact that the work is innovative, one should look critically at several issues:The group was too small and only men were examined. A group of women should also be examined in the future.The lack of analysis of data such as body weight, BMI, stimulants, drugs and the level of daily physical activity. This would certainly allow a more accurate analysis of the effects of the training used.

## 5. Conclusions

Training with the use of a suspension system had a positive effect on the level of exercise tolerance as assessed by the exercise test in the second stage of cardiac rehabilitation.Similar to a standard rehabilitation program, training with the use of a suspension system brought positive changes in selected left ventricular indicators, mainly in the ejection fraction.Resistance training with the use of a suspension system was more effective in improving blood lipid profile compared to standard training.

## Figures and Tables

**Figure 1 ijerph-17-05419-f001:**
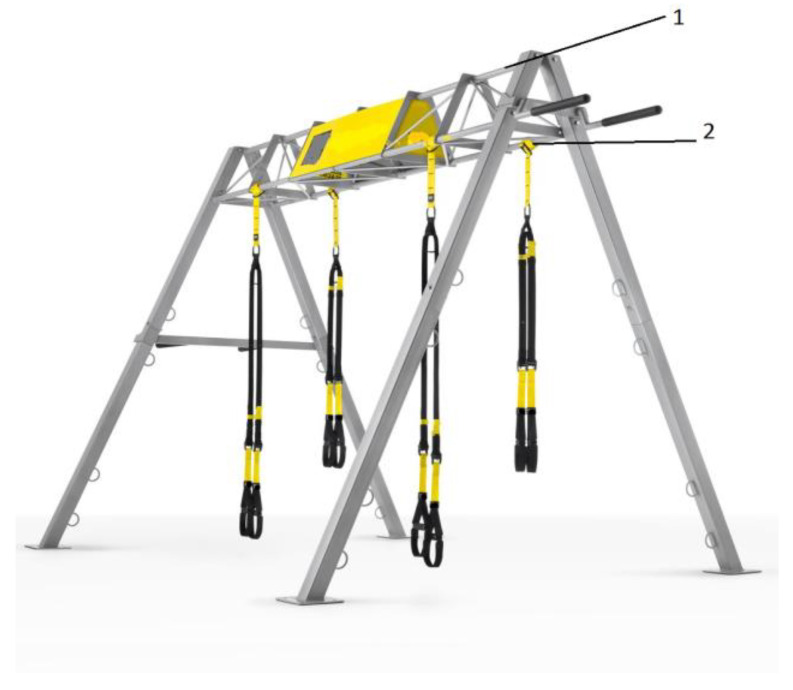
Overall layout of the suspension system platform: (1) Platform, (2) fastening point of the straps.

**Figure 2 ijerph-17-05419-f002:**
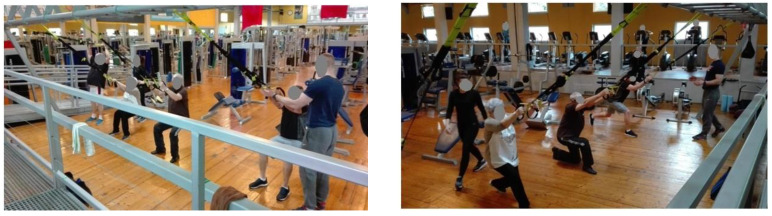
Group exercises with the use of a suspension system.

**Figure 3 ijerph-17-05419-f003:**
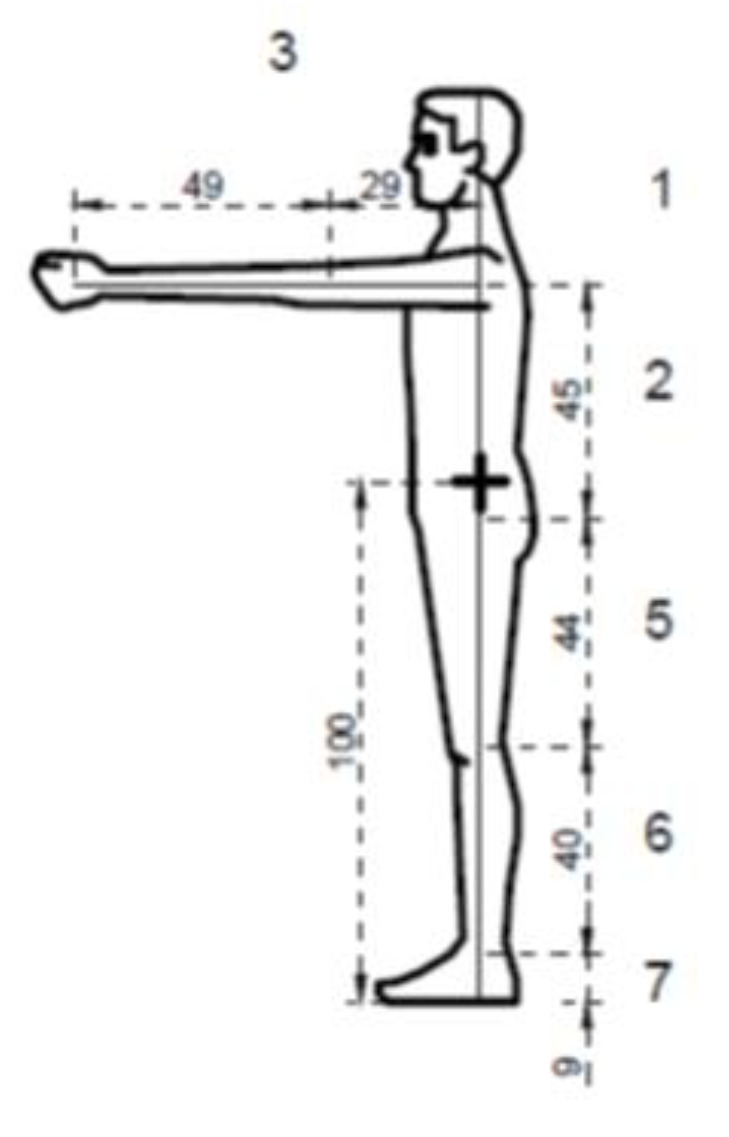
Breakdown into sections of the body.

**Figure 4 ijerph-17-05419-f004:**
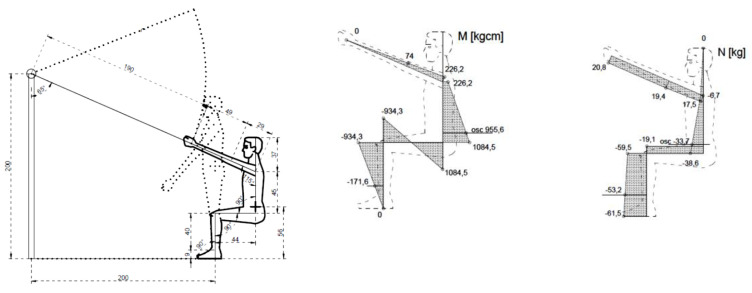
Exercise 1: Bending moment, axial force.

**Figure 5 ijerph-17-05419-f005:**
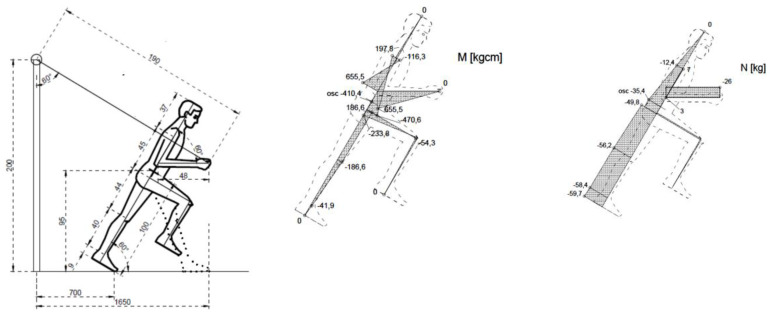
Exercise 2: Bending moment, axial force.

**Figure 6 ijerph-17-05419-f006:**
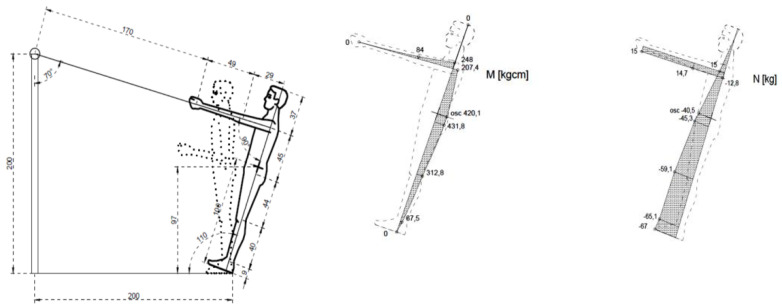
Exercise 3: Bending moment, axial force.

**Figure 7 ijerph-17-05419-f007:**
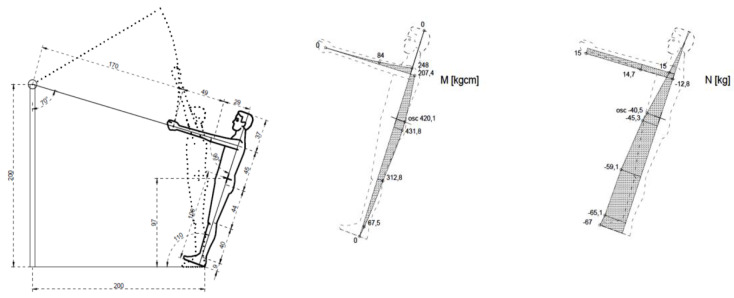
Exercise 4: Bending moment, axial force.

**Table 1 ijerph-17-05419-t001:** Training following ESC recommendations.

Training Type	Methodology	Load
**Endurance training**	Training on a stationary bicycle, 5 times a week for 30 min.	The load applied on the basis of the calculated training heart rate starting with 60% of the heart rate reserve and increasing it by 10% after 5 training units, up to 80% of the heart rate reserve, up to 14th degree of the Borg scale for perceived exertion
**General** **stamina training**	Gymnasium exercises–elements of aerobic and anaerobic training, stretching, breathing exercises general stamina training in a gym with steppers, gym balls, mattresses, and wooden sticks (150 cm), 5 times a week for 30 min.	
**Resistance training**	Resistance training performed for 30 min in a strength training room with the use of elliptical trainers, rowing machines and steppers 5 times a week for 30 min.	

**Table 2 ijerph-17-05419-t002:** Age of subjects.

Variable	SP Group	Standard Group	*p*
	x ± SD	x ± SD	
Age (years)	60.92 ± 7.80	57.45 ± 8.12	0.102

**Table 3 ijerph-17-05419-t003:** Body mass and Body Mass Index (BMI) before (I) and after (II) the 24-day rehabilitation cycle.

Variable	Group: SP x ± SD	*p*-Value	Group: Standard x ± SD	*p*-Value	Δ SP vs. Δ Standard *p*-Value
Body mass (kg) I Body mass (kg) II Δ (kg)	78.54 ± 8.22 78.12 ± 6.31 −0.42	0.841	80.10 ± 7.65 79.21 ± 2.21 −0.89	0.732	0.632
BMI (kg/m^2^) I BMI (kg/m^2^) II Δ (kg/m^2^)	27.88 ± 6.62 27.31 ± 8.51 −0.57	0,446	28.12 ± 8.11 28.01 ± 5.51 −0.11	0.524	0.554

**Table 4 ijerph-17-05419-t004:** Types of coexisting conditions.

Condition Type	SP Group	Standard Group
	N (%)	N (%)
Ischemic heart disease	24 (100%)	20 (100%)
Type 2 diabetes	5 (21%)	5 (25%)
Hyperlipidemia	11 (46%)	9 (45%)
Hypertension	15 (63%)	17 (85%)
Myocardial infarction	24 (100%)	20 (100%)

**Table 5 ijerph-17-05419-t005:** Myocardial infarction type.

Location and Type	SP Group	Standard Group
	N (%)	N (%)
NSTEMI	14 (58%)	8 (40%)
STEMI	10 (42%)	12 (60%)
Total	24 (100%)	20 (100%)

NSTEMI–No ST elevation myocardial infarction, STEMI–ST elevation myocardial infarction. NSTEMI was prevalent in the suspension system group of patients, while STEMI was prevalent in the standard group.

**Table 6 ijerph-17-05419-t006:** Applied treatment type.

Method	SP Group	Standard Group
	N (%)	N (%)
PCI + STENT	24 (100%)	20 (100%)
PTCA	0	0
Total	24 (100%)	20 (100%)

PCI–Percutaneous Coronary Intervention, PTCA–Percutaneous Transluminal Coronary Angioplasty.

**Table 7 ijerph-17-05419-t007:** Number of stents implanted.

Number	SP Group	Standard Group
	*N* (%)	*N* (%)
0	0	0
1	17 (71%)	16 (80%)
2	7 (29%)	3 (15%)
3	0	1 (5%)
4 and more	0	0
Total	24 (100%)	20 (100%)

*N*–Amount.

**Table 8 ijerph-17-05419-t008:** General stamina training unit protocol.

Training Session Part	Duration (min)	Borg Scale	Exercise	Position/Goal
**Warm-up**	5	9–10	Lifting shoulders 3 × 10 Rotating hips 2 × 10 Lifting lower limbs 3 × 10	Standing exercises/improvement of breathing patterns and additional warm-up for joints
**Main part**	5–10	12–13	Lifting shoulders with position control3 × 10, lifting lower limbs in an alternating manner 3 × 10	Coordination and balance exercises on Thera-Band balls, sitting position
	10–15	12–14	Lifting lower limbs 3 × 15, Toe stand 3 × 15 Lifting upper limbs 3 × 15, Torso rotation 3 × 10	Standing position on a mat/improving balance and proprioception
	15–20	11–12	Lifting shoulders 3 × 15 Torso rotation Bending and extending forearms 3 × 15	Exercises with a wooden stick/standing position on a mat
**Relaxing exercises**	20–25	9–10	Breathing exercises	Standing position, relaxation of breath and heart rate
**Stretching**	25–30	9	Two upper limb stretching exercises 2 × 60 s, Two lower limb stretching exercises 2 × 60 s	Stretching muscle areas involved in seating on a mat

**Table 9 ijerph-17-05419-t009:** Resistance training unit protocol.

Training Session Part	Duration (min)	Borg Scale	Position/Technique
**Main part**	1–7 8–15 16–22 23–30	12–13 12–14 13–14 11–12	Stationary bicycle with resistance Stepper with resistance Rowing machine with resistance Elliptical trainer with resistance
**Stretching**	5	9	Two upper limb stretching exercises 2 × 60 s Two lower limb stretching exercises 2 × 60 s

**Table 10 ijerph-17-05419-t010:** Protocol of a training unit with a suspension system.

Training Session Part	Duration (min)	Borg Scale	Position/Technique
**Main part**	1–7.5 (60 s effort, 90 s rest) 7.5–15 (60 s effort, 90 s rest) 15–22.5 (60 s effort, 90 s rest) 22.5–30 (60 s effort, 90 s rest)	12–13 12–14 13–14 11–12	Knee bend Lunges Pulling handles to the chest Shoulder lift
**Stretching**	5	9	Two upper limb stretching exercises 2 × 60 s Two lower limb stretching exercises 2 × 60 s

**Table 11 ijerph-17-05419-t011:** Muscle and spine loads during exercises, expressed in kilograms.

Exercise No.	Amount of Forces at the Centre of Gravity		Force Acting on Spine Straightening Muscles *	Force at the Skeleton *
	M (kNcm)	N (kg)	(kg)	(kg)
I	955.6	−33.7	191.1	−224.8
II	−410.4	−35.4	N/A	46.7
III	420.1	−40.5	84	−124.5
IV	420.1	−40.5	84	−124.5

* Positive values: Expansion, negative values: Compression.

**Table 12 ijerph-17-05419-t012:** Preferred number of repetitions depending on the muscle strength training method used.

Goal of Training	Preferred Number of Repetitions
Maximum force	1–5
Muscle power	4–5
Muscular hypertrophy	6–10
Muscle definition	Min 10–12
Strength	15>

**Table 13 ijerph-17-05419-t013:** Electrocardiographic exercise test results before (I) and after (II) the 24-day training cycle.

Variable	Group: SP x ± SD	*p*-Value	Group: Standard x ± SD	*p*-Value	Δ SP vs. Δ Standard *p*-Value
Duration I Duration II Δ (min)	7.60 ± 1.16 8.71 ± 0.92 1.12	<0.001	7.58 ± 1.10 9.57 ± 1.77 1.99	<0.001	0.460
Distance I Distance II Δ (m)	304.75 ± 56.88 418.73 ± 45.76 113.98	<0.001	301.73 ± 62.48 424.15 ± 110.25 122.42	<0.001	0.346
MET I MET II Δ MET	8.93 ± 1.22 9.96 ± 0.96 1.03	<0.001	9.99 ± 1.08 11.91 ± 1.86 1.925	<0.001	<0.001
VO_2_max I VO_2_max II Δ (ml/kg/min)	34.34 ± 4.95 39.43 ± 4.48 5.09	<0.001	34.21 ± 5.07 44.53 ± 9.25 10.33	<0.001	0.302
HR rest I HR rest II Δ (bpm)	69.58 ± 6.96 68.54 ± 11.44 −1.04	0.637	70.10 ± 8.08 66.25 ± 8.55 −3.85	0.013	0.915
HRmax I HRmax II Δ (bpm)	120.92 ± 12.38 121.00 ± 12.19 0.08	0.974	122.35 ± 15.16 124.65 ± 13.54 2.30	0.379	0.751
SBPrest I SBPrest II Δ (mmHg)	122.92 ± 13.10 114.17 ± 10.60 −8.75	0.013	129.75 ± 14.00 122.50 ± 6.39 −7.25	0.016	0.020
DBPrest I DBPrest II Δ (mmHg)	79.79 ± 6.67 77.71 ± 5.89 −2.08	0.105	78.75 ± 6.46 77.50 ± 6.39 −1.25	0.449	0.926
SBPmax I SBPmax II Δ (mmHg)	155.00 ± 18.42 148.96 ± 10.83 −6.04	0.035	150.25 ± 15.93 146.50 ± 10.89 −3.75	0.292	0.629
DBPmax I DBPmax II Δ (mmHg)	81.67 ± 6.37 80.21 ± 7.59 −1.46	0.283	81.25 ± 7.23 80.00 ± 4.59 −1.25	0.489	0.983

All data is presented as mean values ± standard deviations and difference (Δ–delta), p–Statistically significant level (the lowest assumed level was *p* ≤ 0.05), MET–Metabolic equivalent, VO_2_max–Maximal Oxygen Uptake, HRrest–Heart rate at rest, HRmax–Maximum heart rate, SBPrest–Systolic blood pressure at rest, SBPmax–Maximum systolic blood pressure, DBPrest–Diastolic blood pressure at rest, DBPmax–Maximum diastolic blood pressure.

**Table 14 ijerph-17-05419-t014:** Results of echocardiographic tests carried out before (I) and after (II) the 24-day rehabilitation cycle.

Variable	Group: SP x ± SD	*p*-Value	Group: Standard x ± SD	*p*-Value	Δ SP vs. Δ Standard *p*-Value
LVEDD I LVEDD II Δ (mm)	49.29 ± 3.77 49.08 ± 3.74 −0.21	0.618	50.35 ± 5.11 50.05 ± 4.85 −0.30	0.580	0.718
LVESD I LVESD II Δ (mm)	35.38 ± 3.90 35.92 ± 3.12 0.54	0.264	33.90 ± 3.88 34.65 ± 4.26 0.75	0.183	0.441
LVESV I LVESV II Δ (ml)	47.42 ± 8.07 48.17 ± 5.15 0.75	0.502	48.03 ± 13.21 50.78 ± 15.19 2.75	0.157	0.877
LVEDV I LVEDV II Δ (ml)	126.71 ± 21.69 125.88 ± 20.10 −0.83	0.078	121.85 ± 28.53 120.03 ± 27.00 −1.82	0.530	0.748
LVSV I LVSV II Δ (ml)	79.29 ± 22.68 77.70 ± 22.09 −1.58	0.257	73.80 ± 22.17 69.25 ± 19.53 −4.55	0.250	0.521
LVEF% I LVEF% II Δ (%)	52.29 ± 3.29 53.33 ± 3.27 1.04	0.001	53.30 ± 3.06 55.30 ± 4.13 2.00	0.005	0.321
LVM I LVM II Δ (g)	188.01 ± 35.14 186.63 ± 33.60 −1.38	0.616	203.79 ± 50.30 201.14 ± 46.09 −2.64	0.477	0.427
LVMI I LVMI II Δ (g/m^2^)	94.29 ± 7.94 94.13 ± 6.85 −0.16	0.301	100.71 ± 19.58 100.14 ± 20.20 −0.57	0.198	0.688

LVEDD–Left Ventricular End-Diastolic Diameter, LVESD–Left Ventricular End-Systolic Diameter, LVESV–Left Ventricular End-Systolic Volume, LVEDV–Left Ventricular End-Diastolic Volume, LVSV–Left Ventricular Stroke Volume, LVEF%–Left Ventricular Ejection Fraction, LVM–Left Ventricular Mass, LVMI–Left Ventricular Mass Index.

**Table 15 ijerph-17-05419-t015:** Results of blood lipid profile tests carried out before (I) and after (II) the 24-day rehabilitation cycle.

Variable	Group: SP x ± SD	*p*-Value	Group: Standard x ± SD	*p*-Value	Δ SP vs. Δ Standard *p*-Value
TC I TC II Δ (mg/dL)	180.92 ± 6.30 178.58 ± 5.36 −2.34	0.003	165.00 ± 32.69 157.77 ± 29.92 −7.23	0.107	0.014
HDL I HDL II Δ (mg/dL)	49.13 ± 4.62 50.83 ± 4.32 1.71	<0.001	50.10 ± 14.40 50.13 ± 13.33 0.03	0.991	0.998
LDL I LDL II Δ (mg/dL)	88.04 ± 3.83 85.92 ± 4.32 −2.13	0.005	90.70 ± 26.58 85.50 ± 25.89 −5.20	0.103	0.975
TG I TG II Δ (mg/dL)	114.58 ± 10.31 106.79 ± 23.45 −7.79	0.072	121.06 ± 50.25 110.80 ± 50.36 −10.26	0.245	0.872

TC–Total cholesterol, HDL–High-density lipoproteins, LDL–Low-density lipoproteins, TG–Triglycerides.

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
