# Peer review of "Effectiveness of Resistance Training with the Use of a Suspension System in Patients after Myocardial Infarction"

_ijerph, 2020, doi:10.3390/ijerph17155419_

Round 1

Reviewer 1 Report

The small n makes this more of an exploratory study. The gist of the suspension system can be better explained in the introduction. Some specific comments include:

L 35 “the is”

L 40 Suspension system needs much better explanation

L 46 “bases” should be “is based”

L 49 “bases” should be “is based”

L 75 Awkward wording

L 71-75 Most relevant question, particularly since study design employs a control group would be whether suspension system imparts better improvements (in the various dependent variables) relative to standard training

L 79 Should spell out 44 at start of sentence

L 80 How was randomization done?

L 129-143 Sentence structure awkward. Perhaps bullets needed.

L 157 Table split btw two pages. Also need reformatting as columns not lined up

L 239 Table split btw two pages

Author Response

Thank you very much for providing valuable tips and comments. All corrections were made in the manuscript text.

Reviewer 2 Report

This manuscript reports the findings of a study comparing the use of a suspension system compared with traditional resistance training on changes in exercise testing and echocardiographic measurements in patients recovering from myocardial infarction.  The results indicated that exercise test duration, distance covered, metabolic equivalent, maximal oxygen consumption, and resting systolic blood pressure were significantly different from pre- to post-training in both groups.  Similarly, left ventricular ejection fraction improved in both groups following training.  The suspension system training group had significant improvements in blood total cholesterol, high-density lipoprotein, and low-density lipoprotein.  These findings led the authors to conclude that training with the suspension system had a positive effect on the change of exercise tolerance level, LV function, and blood lipid profile.  The experimental design and general conduct of the study appear sound, however, there is concern regarding the statistical analysis, and some other limitations and data presentations factors should be considered.

Major Comments:

  • The authors state that parametric t-test or non-parametric Wilcoxon paired order tests were performed for dependent variables; however the experimental design (pre-training vs post-training, and suspension system vs. standard) appears to be better suited for a 2x2 split plot ANOVA (pre- vs post training and suspension system vs. standard), which would better control for multiple comparisons as opposed to conducting multiple t-tests. Suggest consultation with a biostatistician to ensure the analysis method is appropriate; also suggest providing rational why between-group differences in pre- to post- training were analyzed by calculating deltas and comparing the deltas with a t-test, rather than a 2x2 split plot ANOVA
  • Was this designed as a superiority or non-inferiority trial comparing the suspension system to standard rehabilitation? It appears to be the latter, but it is unclear and not explicitly stated whether the authors hypothesized that the suspension system would perform similarly or superior to standard treatment.  Again, ensure the appropriate analysis design (e.g., multiple t-tests vs. 2x2 split plot ANOVA) is critical in being able to answer this question
  • This study only included male patients; please provide a rationale for the exclusion of women. Recent guidance from federal funding agencies, such as the National Institutes of Health, requires the inclusion of both male and female research subjects in clinical and pre-clinical studies because of the importance of sex as a biological variable, unless there is a valid physiological or medical reason to select subjects of only one sex.  Please refer to NIH Policies on inclusion of women in clinical research and sex as a biological variable

Minor Comments:

Introduction: Suggest re-organizing into multiple paragraphs instead of a single long paragraph, to improve readability

Lines 54-56: what is meant by “the forced natural and ergonomic position during this training is its fundamental principle, and the benefit to patients at the same time”?

Introduction: More detail on how the physiologic stimuli from the suspension system differs from traditional resistance training, and how the resulting training adaption differs would be useful.  The authors describe “parameters such as trunk stability and proprioception are significantly improved by stimulating the neuromuscular system” but do not differentiate in how the suspension system provides a unique stimulus compared to traditional training in this regard; this will help provide a physiologic basis for the rationale of why the suspension system might be a useful rehabilitation tool

Methods: Suggest consolidating Figures 3-14; for example, Figures 3, 7, and 11 can all be consolidated into a single 3-panel figure, as could Figures 4, 8, and 12, etc.  This would also enable the reader to see the Exercise, Bending Moment, and Axial force diagrams for each exercise in a single pane, rather than having to scroll up and down through all these figures

Methods: The numbers in Figures 3-14 are very small and difficult to read, and in some cases the color (i.e., bright green and yellow) is too light to read; suggest revising to make these easier to see

Methods: Please provide details on blood collection, processing, and analysis, and when these measures were collected

Methods: Were suspension systems exercises done in a group setting?  If so, were standard exercises done in a group setting as well or individually?  If one intervention was a group setting and the other was not, this might be a confounding difference that deserves discussion

Methods: Please provide justification for sample size (for example, was an a priori power analysis conducted?)

Methods: was training load and/or intensity between the suspension group and standard group matched?  If so, how was this measured/quantified?

Results: Can additional patient data (height, weight/body mass, body mass index, smoker/non-smoker, etc) be included to better describe the patient population?  Similarly, were there differences between groups in body mass change from pre- to post-training?

Results: Were there differences in the types of medications that patients in each group were chronically and/or acutely taking?

Results: The data presentation is cumbersome and tables do not appear to be formatted correctly (for example, in several instances “II” appears to have been pushed to the line below); it is also not stated whether +/- values are SD, SEM, or something else; Table 11 – final column comparing delta SP vs delta standard – is this a P-value?  Also, P-values should not be 0.000 (or <0.000, as is stated in parts of the discussion such as Line 357 – a P value cannot be less than 0); this is likely an artifact of the statistical software rounding off the value returned; suggest revising to P < 0.001

Results: Suggest revising Blood Pressure abbreviations to “SBP” and “DBP” (e.g., SBPrest, SBPmax, DBPrest, DBPmax, etc.) as these are more commonly conventionally used rather than “RRsysrest”, which is difficult to remember.  Also please include Mean Arterial Pressure calculations

Discussion: attributing the difference in systolic blood pressure at max to random occurrence seems like a simplistic explanation; could there be a physiologic mechanism that could account for this?  For example, improvements in exercise-mediated local systemic vasodilation in exercising muscle? Or changes in cardiac function/performance?  Suggest critically thinking about this topic and, if the authors still believe this to be random chance, then providing a more detailed rationale of why that is likely to be the correct explanation

Discussion: were changes in LVEF due to changes in LV end-systolic volume, end-diastolic volume or both?  Were there differences in contraction time?

Discussion: potential confounders to the current study, such as differences in medications (for example, hypertension medications, anticoagulant treatment, hemodynamic or inotropic support, etc.) should be discussed

Discussion: a section on the limitations of this study should be included

Conflict of Interest disclosure statement missing

Author Response

Thank you very much for providing valuable tips and comments. All corrections were made in the manuscript text.

I will refer only to the commentary on statistics: The tests performed in the study were selected for demarcation in-group and between-group differences (parametric t-test and Wilcoxon paired order test were used for in-group differences and parametric t-test and U Mann-Whitney test performed for deltas were used for between-group differences). Furthermore, deltas were used to specify the dynamic of changes.

Reviewer 3 Report

Rhis is a nice study and should be published.

I have some minor comments:

The statistics of the tables should be explained more in deatils. Especially, mean and stnd do not explaind p = 0 values (overlapping).

In the introduction some general remarks of heart failure should be included 8especially distrinction form coronary heart disease and heart failure

(for example see: GUSKI, Hans; KOGAN, Evgenya A.; SHVALEV, Vadim N.. Etiology and Pathogenesis of Sudden Cardiac Death. Diagnostic Pathology, [S.l.], v. 5, n. 1, july 2019. 

Author Response

(The authors gave the same response as above.)

Round 2

Reviewer 2 Report

Thank you to the authors for providing this detailed revision, which has addressed the majority of my previous comments.  Thank you also for clarifying the statistical analysis in the author's replay.  My primary concern remains that the statistical analysis may have an unacceptably high Type I error rate due to employment of multiple t-tests for between- and within-group comparisons.  Using multiple t-tests (or non-parametric analogs) to detect between- and within-group differences increases the Type I error rate  with each test conducted (for example, if one test was conducted for between-groups and another for withing groups, if the Type I error rate for each test is 5%, then the combined Type I error rate from conducting two tests is now 10%), which is why ANOVA is a preferred method of making multiple comparisons, allowing for control of Type I error rate with multiple comparisons.  Thus, using a split-plot ANOVA for fixed effect (between-subjects) and random effects (within-subjects) factor controls for multiple comparisons so that the Type I error rate remains at 5%.

Minor Comments:

Methods: Please provide justification for sample size (for example, was an a priori power analysis conducted?) - couldn't find this in revised version

Results: Can additional patient data (height, weight/body mass, body mass index, smoker/non-smoker, etc) be included to better describe the patient population?  Similarly, were there differences between groups in body mass change from pre- to post-training? - thank you for including descriptions of height and weight in inclusion criteria, but a table including the above information, or incorporating this information into Table 2 along with age, is necessary to better describe the patient population studied and ensure there were no differences between groups in these important variables

Thank you for consolidating Figures 3-14, this has improved the ease of reading dramatically.  However the numbers in each illustration are still extremely small and difficult to read.  Please increase the font size for the numbers in these graphics

Check figure numbering (for example, should Figure 15 now be Figure 7?)

Author Response

Thank you very much for the valuable comments of the reviewer, according to which we made appropriate corrections. However, we believe that the proposed change in the method of statistical analysis would make sense in a situation where we would analyze the actual effect as such of both forms of training. The purpose of our research was to determine the usefulness of training using the suspension system as an alternative to standard strength and endurance training in cardiac rehabilitation. We are looking for new, more attractive forms of improving cardiological patients, and this is the proposed training. At work, we asked questions whether such training can be used in the rehabilitation of patients after a heart attack and how it can affect changes in the level of exercise tolerance, changes in selected echocardiographic parameters and changes in the lipid profile (delta). We then compared the obtained results with the results of the control group where standard methods were used. And here also changes caused by training were observed. Then we made inter-group comparisons of these changes, thanks to which we obtained answers to research questions. After consulting the statistician, we received assurance that the method of analysis adopted by us is correct and does not raise any objections. However, we think that another method proposed by the reviewer is very interesting and we will certainly want to use it in our next publication.